

# Mitochondrial complex I deficiency leads to the retardation of early embryonic development in *Ndufs4* knockout mice

Mei Wang[1,2,*], Ya-Ping Huang[1,2,*], Han Wu[2,*], Ke Song[3], Cong Wan[2], A-Ni Chi[2], Ya-Mei Xiao[1] and Xiao-Yang Zhao[2]

[1] State Key Laboratory of Developmental Biology of Freshwater Fish, Hunan Normal University, Changsha, China

[2] Department of Developmental Biology, School of Basic Medical Sciences, Southern Medical University, Guangzhou, China

[3] School of Basic Medical Sciences, Southern Medical University, Guangzhou, China

[*] These authors contributed equally to this work.

## ABSTRACT

**Background**. The *NDUFS4* gene encodes an 18-kD subunit of mitochondria complex I, and mutations in this gene lead to the development of a severe neurodegenerative disease called Leigh syndrome (LS) in humans. To investigate the disease phenotypes and molecular mechanisms of Leigh syndrome, the *Ndufs4* knockout (KO) mouse has been widely used as a novel animal model. Because the homozygotes cannot survive beyond child-bearing age, whether *Ndufs4* and mitochondrial complex I influence early embryonic development remains unknown. In our study, we attempted to investigate embryonic development in *Ndufs4* KO mice, which can be regarded as a Leigh disease model and were created through the CRISPR (clustered regularly interspaced short palindromic repeat) and Cas9 (CRISPR associated)-mediated genome editing system. **Methods**. We first designed a single guide RNA (sgRNA) targeting exon 2 of *Ndufs4* to delete the NDUFS4 protein in mouse embryos to mimic Leigh syndrome. Then, we described the phenotypes of our mouse model by forced swimming and the open-field test as well as by assessing other behavioral characteristics. Intracytoplasmic sperm injection (ICSI) was performed to obtain KO embryos to test the influence of NDUFS4 deletion on early embryonic development. **Results**. In this study, we first generated *Ndufs4* KO mice with physical and behavioral phenotypes similar to Leigh syndrome using the CRISPR/Cas9 system. The low developmental rate of KO embryos that were derived from knockout gametes indicated that the absence of NDUFS4 impaired the development of preimplantation embryos. **Discussion**. In this paper, we first obtained *Ndufs4* KO mice that could mimic Leigh syndrome using the CRISPR/Cas9 system. Then, we identified the role of NDUFS4 in early embryonic development, shedding light on its roles in the respiratory chain and fertility. Our model provides a useful tool with which to investigate the function of *Ndufs4*. Although the pathological mechanisms of the disease need to be discovered, it helps to understand the pathogenesis of NDUFS4 deficiency in mice and its effects on human diseases.

Corresponding authors
Ya-Mei Xiao, yameix@hunnu.edu.cn
Xiao-Yang Zhao,
zhaoxiaoyang@smu.edu.cn

## INTRODUCTION

Mitochondria are the major energy-producing organelles in eukaryotes, and the *Ndufs4* encodes an 18-kD subunit of mitochondrial complex I (CI), which is the largest protein assembly of the respiratory chain and forms the major entry-point of electrons into the oxidative phosphorylation system (OXPHOS). Patients with mutations in *NDUFS4* will develop Leigh syndrome, a neurodegenerative disease with onset in infancy or early childhood (*Anderson et al., 2008*; *Lamont et al., 2017*), and possess a CI assembly defect along with a severe abolishment of CI activity (*Petruzzella et al., 2001*). Patients typically fail to thrive and often develop motor coordination dysfunction, episodic vomiting, blindness, hypotonia, seizures, and ataxia. Mutation of the *Ndufs4* gene in mice leads to physical and behavioral symptoms, including hair loss, growth retardation, and an inactive and slow response, causing fatal encephalomyopathy, which can mimic CI disorder and cause a Leigh-like syndrome (*Ingraham et al., 2009*; *Kruse et al., 2008*). Therefore, *Ndufs4* deletion mice are excellent models with which to study the phenotypes and molecular mechanisms of mitochondrial complex I deficiency and Leigh syndrome.

Energy metabolism is involved in numerous biological events, including fertility and embryogenesis. Researchers found that a higher potential for continued embryogenesis and implantation in humans was associated with embryos that developed from oocytes with greater ATP content (*Van Blerkom, Davis & Lee, 1995*). Another report had shown that mitochondrial dysfunction in mouse oocytes results in preimplantation embryo arrest *in vitro* (*Thouas et al., 2004*). Moreover, many aberrations during early mouse embryonic development and the increasing health risks in the offspring may be caused by numerous dysregulated genes related to mitochondrial complex I in IVF (*in vitro* fertilization) embryos (*Ren et al., 2015*). These data indicated that both mitochondria and CI play crucial roles in embryonic development. NDUFS4, a critical subunit of CI whose effect on embryogenesis and fertility remains unknown, is worthy for further research, which could shed light on the relationship between mitochondria and early embryos.

In previous studies, *Ndufs4* KO mice were obtained by TALEN (Transcription activator-like effector nucleases) (*Kruse et al., 2008*) or homologous recombination (*Ingraham et al., 2009*; *Leong et al., 2012*). Recently, the CRISPR/Cas9 system has been considered as a revolutionary genome-editing technique (*Hsu, Lander & Zhang, 2014*; *Mali, Esvelt & Church, 2013a*; *Segal, 2013*). The CRISPR/Cas9 system was first discovered as an RNA-based adaptive immune system in bacteria and archaea, which used CRISPR to identify and Cas proteins to cut exogenous DNA sequences (*Cong et al., 2013*; *Jinek et al., 2012*; *Mali et al., 2013b*; *Wiedenheft, Sternberg & Doudna, 2012*). Traditionally, to generate mice with gene modification, targeted mouse embryonic stem cells (ESCs) were injected into blastocysts to obtain mice with chimeric germ cells. Here, we injected capped polyadenylated Cas9 mRNA and sgRNA into mouse zygotes to generate mutant mice in one step, which was cheaper and time-saving (*Wang et al., 2013*; *Yang, Wang & Jaenisch, 2014*; *Yang et al., 2013*; *Zhou et al., 2014*).

In this study, we first generated *Ndufs4* KO mice with physical and behavioral phenotypes similar to Leigh-like syndrome using the CRISPR/Cas9 system on mouse embryos. We

further found that knockout of *Ndufs4* in mice impaired the embryonic developmental rate during preimplantation stages. Our data revealed that NDUFS4 or mitochondria complex I plays an important role in early embryonic development.

## MATERIALS AND METHODS

### Animal experiments

B6D2F1 mice were bred with C57BL/6 female (purchased from laboratory animal center of Southern Medical University) and DBA2 male (purchased from Nanjing Biomedical Research Institute of Nanjing University) mice at the laboratory animal center of Southern Medical University. ICR mice were purchased from Guangdong Medical Laboratory Animal Center. All experiments were approved by the Southern Medical University ethics committee (00125817).

### RNA synthesis

To make capped polyadenylated Cas9 mRNA, px330 was used as the template DNA to amplify the Cas9 coding sequence, and the T7 promoter was added by using the primers T7-Cas9-F and Cas9-R. The PCR product was purified using the ZYMO gel recovery kit, and then, it was used as the template for *in vitro* transcription using the mMESSAGE mMACHINE SP6 or T3 kit (Invitrogen). For the sgRNA, the T7 promoter was added by PCR amplification using template px330 and the primers T7-sgRNA-F and sgRNA-R. Purified T7-sgRNA PCR products were used as a template for *in vitro* transcription using T7 RNA polymerase (New England Biolabs). After *in vitro* transcription, the capped polyadenylated Cas9 mRNA and sgRNA were purified with the OMEGA microelute RNA clean-up kit.

### Pronuclear microinjection and embryo transfer

Pronuclear microinjection was performed based on a previously reported Nature protocol (*Yang, Wang & Jaenisch, 2014*). Zygotes were collected from the oviducts of hormone-superovulated B6D2F1 female mice that were crossed with B6D2F1 male mice one day before they were sacrificed. Then, a mixture of the aforementioned Cas9 mRNA (100 ng/$\mu$l) and sgRNA (50 ng/$\mu$l) was injected into one of the pronuclei using a micromanipulator system (Narishige) that was equipped with a microinjector (FemtoJet; Eppendorf). After nucleotide injection, we cultured the zygotes to the blastocyst stage *in vitro* and then transferred the embryos into the uteri of recipient pseudopregnant females. Full-term pups were delivered naturally or by cesarean section.

### Genotyping

To verify the genotype, genomic DNA was extracted from the toes of the 7~10-day-old postnatal mice using the OMEGA MicroElute Genomic DNA Kit. The sequence around the target site (536 bp) was amplified using the genomic DNA template and the primers *Ndufs4*-test-F and *Ndufs4*-test-R. Then, the PCR products were cloned into the TA Cloning Vector and subsequently underwent sequencing. The specific primers used in this paper are as follows: T7-Cas9-F, ttaatacgactcactataggGGAGAATGGACTATAAGGACCACGAC;

Cas9-R, GCGAGCTCTAGGAATTCTTAC; T7-sgRNA-F, ttaatacgactcactataggTA TAACAGTTGATGAGAAAC; sgRNA-R, AAAAGCACCGACTCGGTGCC; *Ndu-fs4*-test-F, TACTGTTCAAGCAGCGTGTT; and *Ndufs4*-test-R, ATGGGCTCACATTA CCAC.

## Western blot

Livers were placed into lysis buffer (10 mM Tris–HCl [pH 8.0], 10 mM NaCl, and 0.5% NP-40) containing protease inhibitors (Roche) and homogenized by a shaker. Then, lysates were centrifuged at 12,000 g for 20 min at 4 °C. The supernatants containing loading buffer were boiled for 5 min. Western blotting transfers were carried out in BioRad transblot chambers. After blocking in 5% milk in phosphate-buffered saline Triton X100 (PBST) for 1 h, the membranes were incubated at 4 °C overnight with anti-NDUFS4 mouse antibody (Abcam, ab87399) followed by incubation with an HRP-conjugated secondary antibody for 1 h at room temperature. The signals were visualized with the Super Signal West Femto reagent (Thermo scientific).

## Statistical analyses

For statistical analyses, $F$-tests were performed to test the equality of variances in the data between WT and KO groups. Comparisons of the significant differences between the means of the WT and KO groups were performed by unpaired $t$-tests. Statistical significance was reached when $p < 0.05$. $* p < 0.05$, $** p < 0.01$, $*** p < 0.001$.

## Open-field test

Activity monitoring was performed in a square-shaped, blue open field measuring 75 cm $\times$ 75 cm that was evenly illuminated at 15 lux[50]. Mice were placed facing the center of the walls and were allowed to move in the apparatus; they were monitored for 10 min by a video camera. The resulting data were analyzed using the image processing system TopScan. The following parameters were assessed: total distance moved, velocity, mean distance to the (nearest) wall and time in the center.

## Intracytoplasmic sperm injection

ICSI was performed as previously described (*Zhou et al., 2016*). We broke the tail of the sperms isolated from the cauda epididymis using an ultrasonic cell disruptor, and superovulated MII oocytes were collected as described above. The sperms were exposed to 5 mg/ml cytochalasin B in M2 medium, and individual sperm was injected into mature oocytes with a Piezo-driven pipette followed by culture in KOSM medium to the blastocyst stage.

## RESULTS

### Generation of the *Ndufs4* KO mouse model by CRISPR

The NDUFS4 protein includes two parts, the transit peptide and the NDUFS4 mature protein. NDUFS4 is encoded by five exons, among which exon 2 encodes the first 17 amino acids of the NDUFS4 protein and the last part of the transit peptide, which is important for targeting on mitochondria (*Breuer et al., 2013*). To generate *Ndufs4* KO mice by CRISPR (Fig. 1A), we designed the sgRNA to target exon 2 of the *Ndufs4* gene

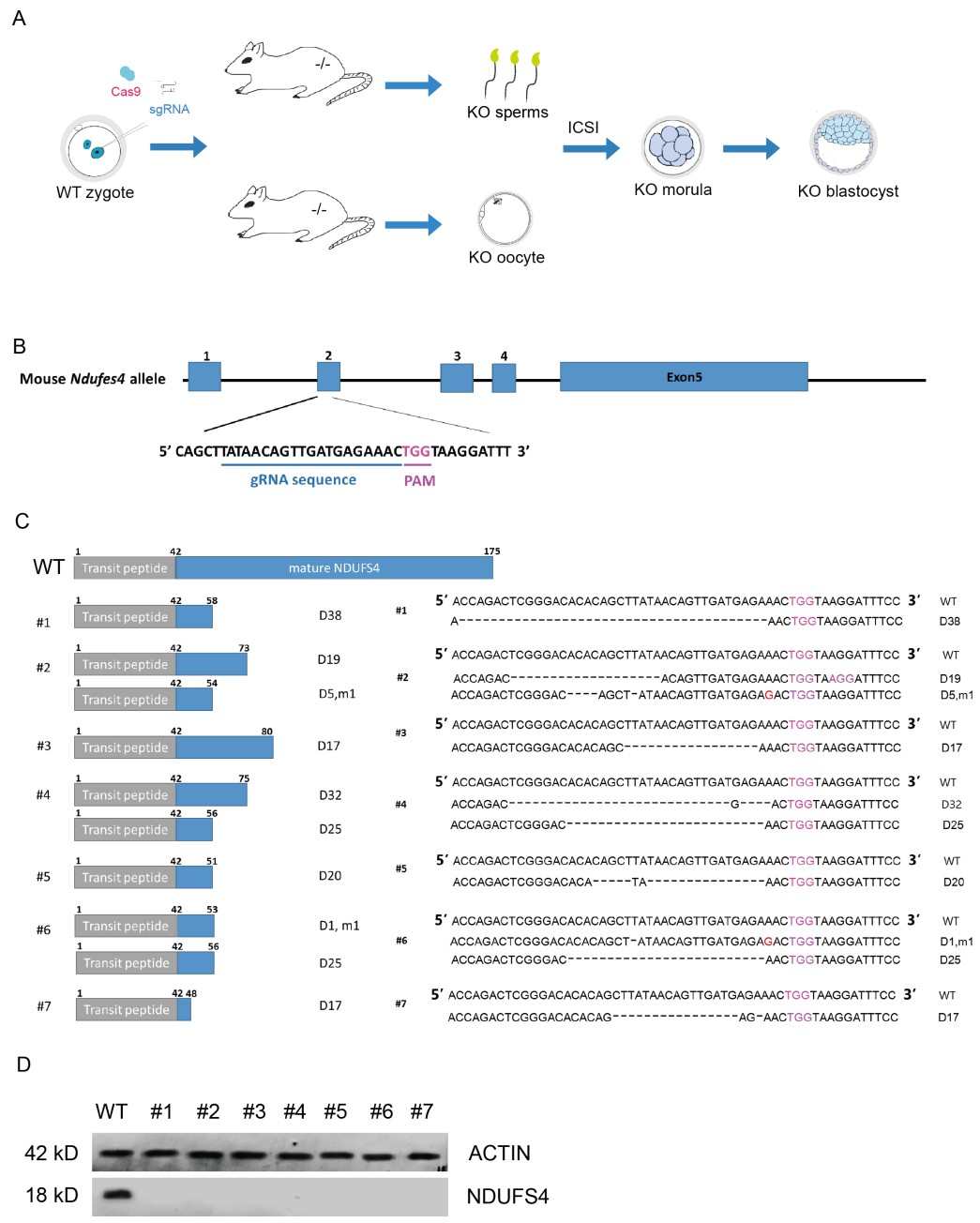

**Figure 1  Generation of *Ndufs4* KO mouse model.** (A) Strategy for generating the *Ndufs4* KO mouse model, and the procedure of our study. (B) Schematic of the Cas9/sgRNA-targeting sites in the *Ndufs4* genomic locus. The guide RNA sequence is marked by a blue line, and the protospacer-adjacent motif (PAM) sequence is labeled in rose red. (C) The NDUFS4 protein is truncated in *Ndufs4*$^{-/-}$ mice, and the right panel shows the corresponding sequence. (D) Western blot experiments showed that the NDUFS4 protein was completely abolished in the *Ndufs4*$^{-/-}$ mice.

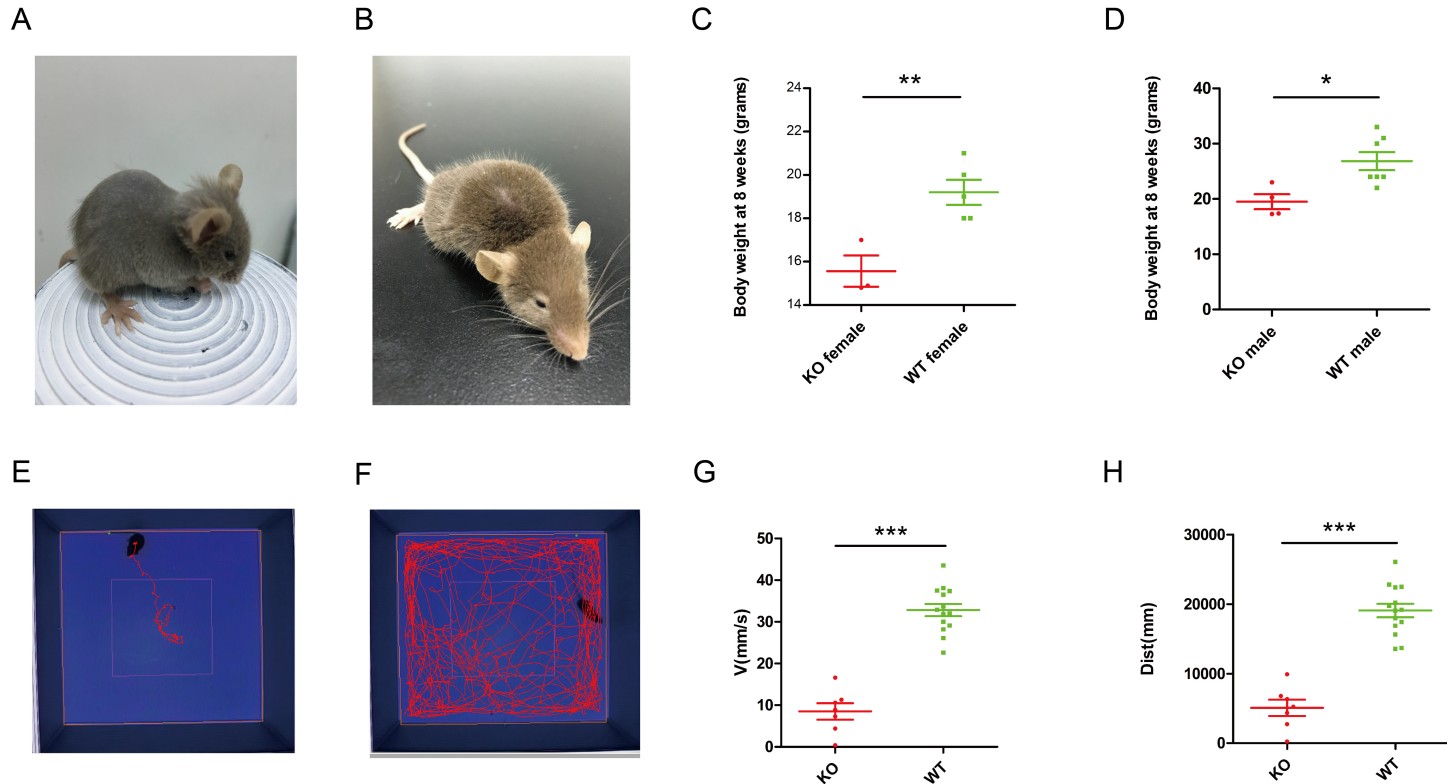

**Figure 2  Phenotype of *Ndufs4* KO mice.** *Ndufs4* KO mice lose their body hair at approximately 3 weeks old (A), and the characteristic would diminish along with growth (B). Both male (C) and female (D) *Ndufs4* KO mice significantly lose weight at 8 weeks compared to WT mice. $*p < 0.05$ and $**p < 0.01$. The motion tracker picture showing the spatial pattern of *Ndufs4* KO mice (E) and WT (F) mice in the open field; KO mice walked significantly less than WT mice in the field. *Ndufs4* KO mice suffered from the rapid deterioration of motor ability in the open field. KO mice moved significantly slower (G) and less (H) than the WT mice in the open-field test. $***p < 0.001$.

(Fig. 1B). Then, we synthesized capped polyadenylated Cas9 mRNA and sgRNA by *in vitro* transcription and co-injected them into 89 mouse embryos at the pronuclear (PN) stage (Fig. 1A). The embryos were cultured to the blastocyst stage and transplanted into pseudopregnant female mice. Finally, we got 7 homozygous *Ndufs4* KO pups (Fig. 1C). In 20 predicted off-target sites, we did not observe any sense mutations on the coding sequences (Tables S1 and S2) (*Stemmer et al., 2015*; *Yang et al., 2013*; *Zong et al., 2017*). Through sequencing, we found deletions in exon 2, which hindered the synthesis of mature NDUFS4. Previous research showed that transposable element insertion into exon 3 could thoroughly diminish NDUFS4 expression (*Leong et al., 2012*). In our data, NDUFS4 proteins were completely abolished in these KO mice, as shown in Fig. 1D.

## Phenotypes of *Ndufs4* knockout mice

By 3 weeks after birth, all homozygous KO mice had begun to lose their body hair (Fig. 2A). However, their hair grew back during the next hair-growth cycle (Fig. 2B). Also, the body weight of KO mice was significantly lower than that of wild-type (WT) mice at 8 weeks both in females ($n = 3$, $p = 0.0082$) and in males ($n = 4$, $p = 0.00142$) (Figs. 2C, 2D).

To confirm the phenotype of the KO mice, the forced swim test was performed to estimate the motor ability of the *Ndufs4* KO mice. In the forced swim test, WT mice could swim for nearly 20 min, while KO mice sank to the bottom as soon as they got into the water. It seemed that the *Ndufs4* KO mice were extremely frail.

Then, we performed the open-field test to record the total distance traveled and the velocity in the trial area. The open-field test is currently one of the most popular procedures in animal psychology. In fact, it has become a convenient way to measure the activity of animal models (*Prut & Belzung, 2003*). Motion tracker data showed that the movement tracks of 7 KO mice were much less than that of the 14 control mice (Fig. 2E, 2F). Locomotor disturbances with regard to the velocity ($p < 0.0001$) and the distance traveled ($p < 0.0001$) among *Ndufs4* knockout mice were observed compared to control mice in the open field (Figs. 2G, 2H), indicating that *Ndufs4* KO mice suffered from the rapid deterioration of motor ability. Altogether, we verified that the *Ndufs4* KO mice generated using the CRISPR/Cas9 system could mimic Leigh syndrome and function as a disease model.

## NDUFS4 affected early embryonic development in mice

The KO mice became weak and died approximately 6 weeks after birth, and few of them survived beyond 9 weeks, which meant that the KO mice rarely generated offspring during their lifetime. Meanwhile, the KO mice could not produce pups by natural mating with WT mice. To assess the effect of NDUFS4 on embryos, we performed ICSI using KO gametes from mouse #1 (female) and mouse #2 (male). The KO and WT gametes were divided into four groups, and we performed ICSI on nearly 40 embryos in each group (Table 1). After the injection, pronuclei could be observed in each group, which indicated the fertility of *Ndufs4* KO gametes. Subsequently, the embryos progressed from the 2-cell stage to the blastocyst stage *in vitro*. We did not observe significant differences in the developmental rates between embryos generated from KO sperms that were injected into WT oocytes or WT sperms that were injected into KO oocytes compared to the embryos generated from WT sperms that were injected into WT oocytes (Fig. 3A). However, the developmental rate of zygotes derived from KO sperms and KO oocytes was significantly lower than that of WT embryos at the 2-cell stage (78.4% versus 97.5%), 4-cell stage (62.2% versus 92.5%), morula stage (51.4% versus 85%) and blastocyst stage (29.7% versus 70%). The ICSI data demonstrated that *Ndufs4* knockout impaired the preimplantation embryonic developmental ability in mice. The transplantation of 11 KO blastocysts into pseudopregnant mice did not result in any live births. However, 7 offspring were obtained from the transplantation of 28 control blastocysts. On one hand, the transplantation data may be a result of the limited KO embryos, but on the other hand, it suggested that the *Ndufs4* knockout blastocysts generated from KO gametes may have an extremely low efficiency to develop to full-term.

These results raised the possibility that abnormalities in the gonads of KO mice may exist. To verify this hypothesis, we checked the testes and ovaries of KO mice. Compared with controls, the ovaries and testes of 8-week-old postnatal KO mice were similar in size (Figs. 3C, 3D). In addition, hematoxylin and eosin (H&E) staining of the ovaries at 8 weeks (Figs. 3D, 3E) showed that the KO ovaries harbored plenty of follicles at different stages and that the number of follicles in KO ovaries (79 and 76 oocytes in 2 KO mice) was much

**Table 1  Development of ICSI embryos from *Ndufs4* KO mice.**

| Background | No. of injected oocytes | No. of 2-cell (% injected) | No. of 4-cell (% injected) | No. of morula (% injected) | No. of blastocyst (% injected) |
|---|---|---|---|---|---|
| KO oocytes x WT sperms | 39 | 38(97.4) | 36(92.3) | 32(82.1) | 24(61.5) |
| KO oocytes x KO sperms | 37 | 29(78.4) | 23(62.2) | 19(51.4) | 11(29.7) |
| WT oocytes x KO sperms | 43 | 38(88.4) | 37(86.0) | 37(86) | 28(65.1) |
| WT oocytes x WT sperms | 40 | 39(97.5) | 37(92.5) | 34(85) | 28(70) |

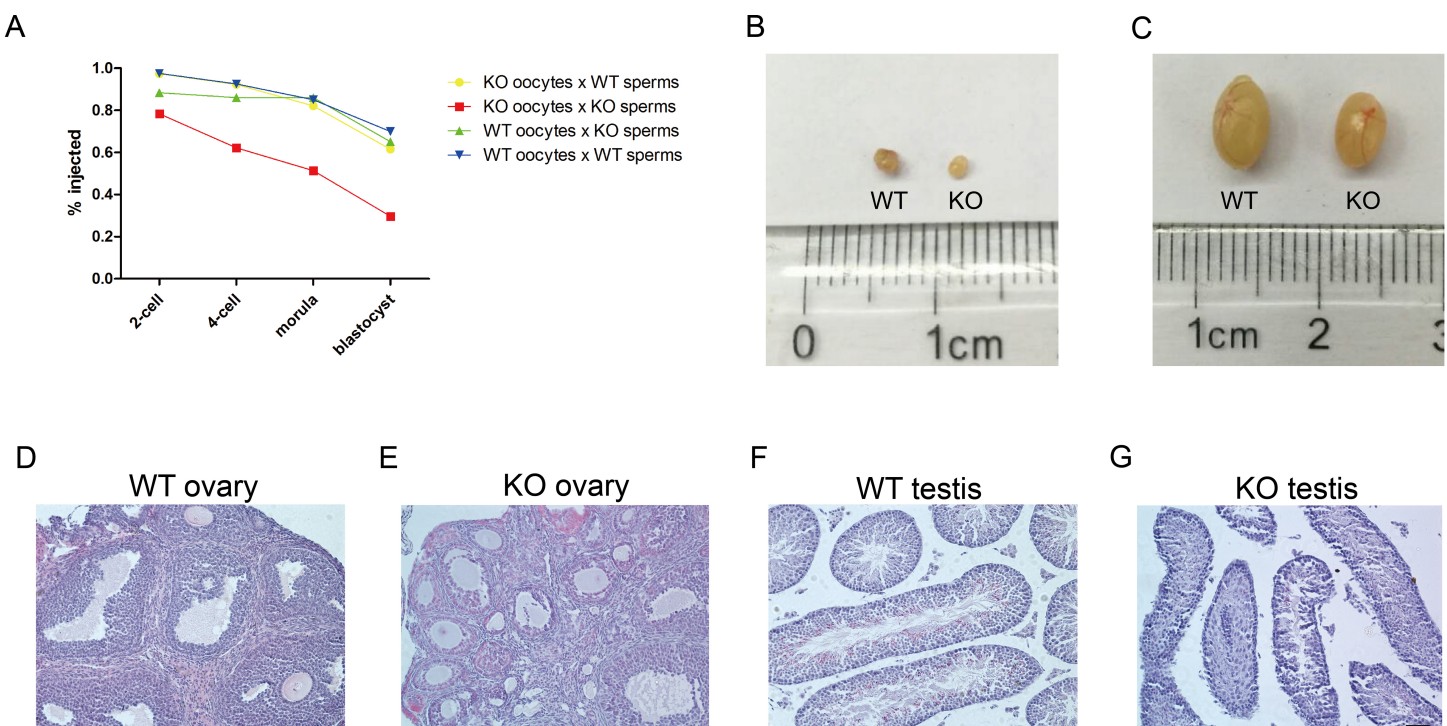

**Figure 3  NDUFS4 affected the early embryos development in mice.** (A) Development of ICSI embryos from *Ndufs4* KO mice in the broken line graph that refers to Table 1. The ovaries (B) and testes (C) of WT and *Ndufs4* KO mice. (D–G) HE staining of the ovaries and testes in KO and WT mice at 8 weeks old. The KO ovary contained plenty of follicles, which were more abundant than that in the WT ovary. Intact seminiferous epithelium and many mature elongated spermatozoa could be found in the KO seminiferous tubes versus WT testis. Scale bar, 50 μm.

greater than that in the WT ovaries (36 and 45 oocytes in 2 WT mice). It was consistent with our findings that the MII oocytes obtained from mouse #1 (80 oocytes) were much more abundant than the oocytes that were obtained from a WT female mouse (40~50 oocytes) after superovulation. This result indicated that there was a spontaneous ovulation arrest in KO female mice. However, the details of this mechanism are not clear. The KO testes contained intact seminiferous epithelium and mature elongated spermatozoa in the seminiferous tubes (Figs. 3F, 3G).

Taken together, these data demonstrated that NDUFS4 plays a role in early embryonic development and spontaneous ovulation in mice.

## DICUSSION

Leigh syndrome is a progressive neurodegenerative disorder that is caused by mitochondrial oxidative phosphorylation defects. The cause of the syndrome is complex, including mitochondrial DNA point mutations and some respiratory chain enzyme defects, such as complex I, IV and pyruvate dehydrogenase. In addition, complex I deficiency is more common than previously recognized (*Lamont et al., 2017*; *Rahman et al., 1996*). Though Leigh syndrome is a genetically heterogeneous entity (*Gerards, 2014*), all of the biochemical defects described to date in patients with LS affect terminal oxidative metabolism and are likely to impair energy production. Typically, OXPHOS dysfunction mostly affects cell types that heavily rely on mitochondrial ATP generation.

Preimplantation mouse embryonic development includes the processes of zygote cleavage to blastocysts, especially from the morulae stage to the blastocyst stage, which involves a series of high energy consumption events. Previous reports have suggested that low ATP was not only associated with mouse MII oocyte spindle impairment (*Zhang et al., 2006*) but also associated with a reduction in the quality of the developed embryos (*Van Blerkom, 2004*). In our study, the KO zygotes had a worse embryonic cleavage rate than that of heterozygous and WT embryos. It indicated that the low developmental rate may be due to *Ndufs4* knockout. Consequently, there may be some correlation of low ATP and early embryos retardation with the knockout of *Ndufs4*. However, we could generate homozygous mutant mice by Cas9 and sgRNA mRNA injection into WT zygotes at a relatively normal frequency, which may have been due to the normal NDUFS4 protein sustained from WT zygotes.

In 2013, Federica Valsecchi et al. found that the primary fibroblasts of $Ndufs4^{-/-}$ mice displayed increased ROS levels and aberrant mitochondrial morphology (*Valsecchi et al., 2013*). ROS cannot only alter most types of cellular molecules but also induce developmental block and retardation (*Guerin, El Mouatassim & Menezo, 2001*). However, whether the ROS accumulation and aberrant mitochondrial morphology in *Ndufs4* KO embryos led to retardation remains to be determined.

In mammals, CI requires the correct assembly of 45 subunits encoded by both nuclear and mitochondrial DNA in order to function correctly (*Koopman et al., 2010*), and the structure is evolutionary conserved (*Leong et al., 2012*). Because of the complexity of CI, the deletion of the other units of CI may also lead to the same result as *Ndufs4* KO or influence the function of CI. Transcriptome profile data showed that there were many mitochondria-related genes that were differentially expressed between *in vitro* fertilization (IVF) and *in vivo* fertilization (IVO) embryos, including *NDUFS1*, *NDUFV1*, *NDUFA2*, *NDUFB8* and *NDUFA9*, which are subunits of CI together with *NDUFS4* (*Ren et al., 2015*). The abnormal expression of these genes led to dysfunction of the mitochondria and, subsequently, IVF-induced embryonic defects. Notwithstanding, the role of NDUFS4 and other CI subunits in embryonic development also needs to be further investigated.

In addition, mutations of other genes encoding enzymes related to the respiratory chain lead to early embryonic death. *Surf1* encodes one of the assembly proteins involved in the formation of cytochrome c oxidase (COX); mutations in this gene also contributed to

the phenotype of Leigh-like syndrome, and constitutive knockout of *Surf1* was associated with increased embryonic lethality (*Agostino et al., 2003*). Knockout of *Slc25a19* caused mitochondrial thiamine pyrophosphate depletion and embryonic lethality. Although the reason behind embryonic retardation is unknown, these data may provide ideas for us to investigate the mechanisms in *Ndufs4* KO embryos.

## CONCLUSIONS

In summary, *Ndufs4* KO mice were first obtained using the CRISPR-Cas9 system, which is a more efficient and time-saving option for generating genetically modified animals than that used in previous studies. We not only observed hair loss and weight loss but also motor impairment in *Ndufs4* KO mice. A role for NDUFS4 in early embryonic development and ovulation was indicated, shedding light on its roles in the respiratory chain and fertility. Moreover, our model provided a useful tool with which to investigate the function of *Ndufs4*, thus helping to understand the pathogenesis of NDUFS4 deficiency.

## ACKNOWLEDGEMENTS

We thank Zhao-Ting Liu, Yu-Jia Shi and Jing Wang for reviewing the manuscript.

### Funding

This work was supported by the National Natural Science Foundation of China (31671544), Hunan Provincial Innovation Foundation of Postgraduate (CX2014B228), and the Ministry of Science and Technology of China (2016YFC1000606). The funders had no role in study design, data collection and analysis, decision to publish, or preparation of the manuscript.

### Grant Disclosures

The following grant information was disclosed by the authors:
National Natural Science Foundation of China: 31671544.
Hunan Provincial Innovation Foundation of Postgraduate: CX2014B228.
Ministry of Science and Technology of China: 2016YFC1000606.

### Competing Interests

The authors declare there are no competing interests.

### Author Contributions

- Mei Wang and Ya-Ping Huang conceived and designed the experiments, performed the experiments, analyzed the data, contributed reagents/materials/analysis tools, wrote the paper, prepared figures and/or tables, reviewed drafts of the paper.
- Han Wu conceived and designed the experiments, performed the experiments, analyzed the data, contributed reagents/materials/analysis tools, prepared figures and/or tables.
- Ke Song, Cong Wan and A-Ni Chi performed the experiments, contributed reagents/materials/analysis tools.

- Ya-Mei Xiao and Xiao-Yang Zhao conceived and designed the experiments, analyzed the data, wrote the paper, reviewed drafts of the paper.

## Animal Ethics

The following information was supplied relating to ethical approvals (i.e., approving body and any reference numbers):

Southern Medical University ethics committee provided full approval for this purely observational research (00125817).

## Data Availability

The raw data has been supplied as a Supplementary File.

## Supplemental Information

Supplemental information for this article can be found online at http://dx.doi.org/10.7717/peerj.3339#supplemental-information.

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
