# Peer review of "Mitochondrial complex I deficiency leads to the retardation of early embryonic development in Ndufs4 knockout mice"

_PeerJ, doi:10.7717/peerj.3339_

## Round 0.1 · original submission · Minor Revisions

Please revise your manuscript according to the reviewers' suggestion. Statistical analysis should be included in Materials and Methods section. Please add the quantifications of numbers of follicles per ovary, and others.

Reviewer 1 ·

Basic reporting

No comment

Experimental design

No comment

Validity of the findings

No comment

Additional comments

In this study, the authors investigated the role of NDUFS4 during mouse early embryo development by CRISPR /Cas9 mediated genome editing system. They showed the Ndufs4-KO mice with the physical and behavioral phenotypes similar to Leigh-like syndrome. Specifically, NDUFS4 absence severely impaired the development of preimplantation embryos, shedding light on the respiration chain and fertility. Also, this model provides a useful tool to understand the pathogenesis of NDUFS4 deficiency in mice and related human diseases.

The manuscript is well written with clear rationale and conclusions based on the results presented, also will be of significant interest to the scientific community.

Some questions that this reviewer has:

1) Table 1, statistical analysis should be indicated in Table.

2) Figure 3, Scale bar needs to added in Fig 3C for ovary section;

3) Is the reduced size of ovary and testis of KO mouse possibly related to its significantly decreased body weight, as shown in Fig 1?

4) For Materials and Methods section, statistical analysis should be included.

5) Some sentences need to be double checked for grammar mistakes, for example, page 11 line 232, “These results raised a possibility that there is abnormal in genitals of KO mice” .

·

Basic reporting

In present study the authors generated Ndufs4 KO mice and investigated the possible effects of Ndufs4 deletion on embryo development. Overall, the authors presented clearly about their results, and provided sufficient background introduction. And the results could support their conclusions.

Experimental design

The authors adopted the widely used CRISPR/Cas9 system to generate KO mice; this approach is widely accepted to study the functions of genes in different models.

Validity of the findings

The authors successfully generated the KO mice and the preliminary results were observed. The results will provide the basic information about the roles of Ndufs4 on early embryo development.

Additional comments

This reviewer has the following suggestions to improve the manuscript:
1. L110, The introduction part the authors claimed that NDUFS4 and mitochondria complex I play roles, this interpretation may not be appropriate, since the authors stated that NDUFS4 is just a submit of this complex, "and“ to ”or" "/" will be correct.
2. The authors should re-check the typo like Abstract part "can't", L243 "IV".
3. For the embryo development examination (Table 1), please provide the error bar, the p value analysis for the better understanding to the readers. P224 and other line, please also enrich this information in the Results part (error, p value and n=?)
4. Detailed information missed in the Figure legends. Please enrich the explanation for the figure legends, for example, Figure 3 (C).
5. In the dicussion part, the authors indicated that Ndufs4 may function through mitochondria-related oxidative stress, why the authors did not examine this?

Reviewer 3 ·

Basic reporting

no comment.

Experimental design

no comment.

Validity of the findings

no comment.

Additional comments

In this study, the authors investigated the Ndufs4 KO mice with the physical and behavioral phenotypes similar to Leigh-like syndrome.Loss of Ndufs4 leads to defective early embryonic development. The manuscript by Wang et al is on an interesting topic.There are several modifications I would suggest to be made.
1. The authors need to proof read the entire manuscript carefully, and to make correct statements.
2. The authors need to add the quantifications of numbers of follicles per ovary.

---

## Round 0.2 · Minor Revisions

This manuscript contains a Statistical Analysis section. Please provide the methodological standard in your Materials and Methods.

# Staff Note: We also recommend that you look again at the English language. For example, from the caption to Fig 3. : "KO ovary owned plenty follicles at different stages, which were more than that in WT ovary." (there are many other examples, we recommend you have a native English speaking colleague review it for you).

---

## Round 0.3 · accepted · Accept

This manuscript was revised according the reviewers' suggestion and can be accepted by Peer J now.